# Unilateral Tamoxifen-Induced Retinopathy as a Consequence of Breast Cancer Treatment—Multimodal Imaging Value

**DOI:** 10.3390/diagnostics13071250

**Published:** 2023-03-27

**Authors:** Paulina Szabelska, Katarzyna Paczwa, Joanna Ciszewska, Radosław Różycki, Joanna Gołębiewska

**Affiliations:** 1Department of Ophthalmology, Military Institute of Aviation Medicine, 01-755 Warsaw, Poland; 2OPTIMUM Ophthalmological Medical Center, 00-501 Warsaw, Poland

**Keywords:** tamoxifen-induced retinopathy, maculopathy, tamoxifen, breast cancer, optical coherence tomography, optical coherence tomography angiography, OCT, OCTA, multimodal imaging

## Abstract

Tamoxifen is a drug used in breast cancer therapy, which inhibits the division of neoplastic cells targeting estrogen receptors. The drug is generally well-tolerated and its use does not cause serious side-effects. The standard dose of the drug is 20 mg once a day for 3 to 5 years. Available epidemiological data have shown that the incidence of ocular toxicity of tamoxifen ranges between 0.9% and 12.0% and increases with higher tamoxifen dose. A rare known complication of tamoxifen use is the development of retinopathy. We present a case of 57-year-old woman presented to an ophthalmologist with decreased visual acuity in her right eye. She has been treated with tamoxifen 20 mg daily for 7 years for breast cancer. Clinical examination and multimodal imaging methods help confirm the diagnosis of unilateral tamoxifen associated retinopathy (TAR). Optical coherence tomography angiography (OCTA) was crucial in the diagnostic process and differential diagnosis, especially in differentiating it from type 2 macular telangiectasias. The correct diagnosis of TAR is very important in deciding the treatment option of tamoxifen. Based on our diagnosis, the oncologist recommended another course of treatment. Tamoxifen therapy was discontinued and switched to letrozole 2.5 mg once a day. The patient attends ophthalmological examination regularly. Visual acuity, OCT and OCTA results remain stable.

**Figure 1 diagnostics-13-01250-f001:**
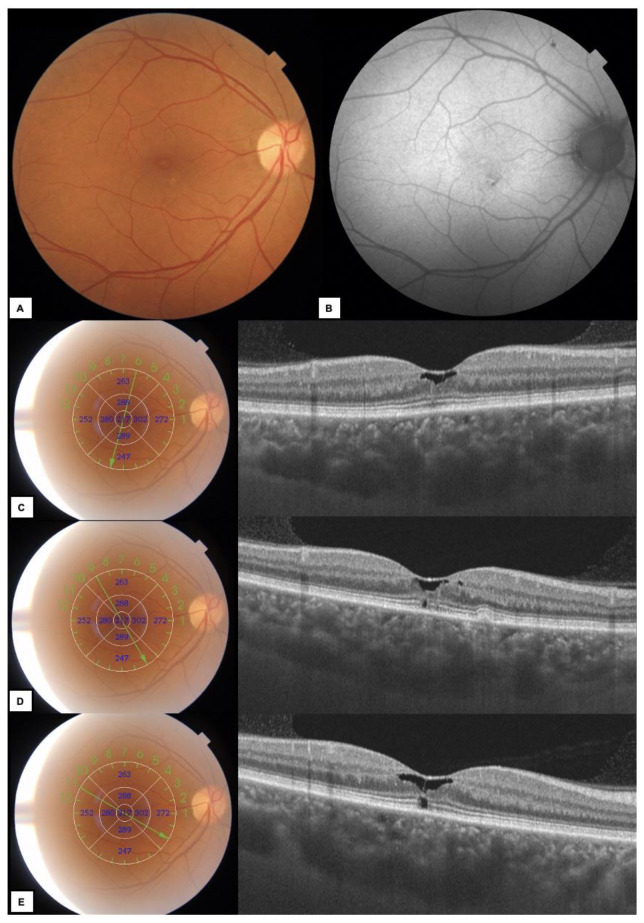
(**A**) Color fundus photography of the right eye (RE) shows subtle alteration of the retinal pigment epithelium (RPE) in the macula. (**B**) Fundus autofluorescence (FAF)–uncharacteristic hypo–and hyperautofluoresence foci in the macula. (**C**–**E**) Different projections of a radial 9 mm optical coherence tomography (OCT) B-scan reveal irregular hyporeflective spaces in the foveal area and photoreceptor’s layer disruption with normal central retinal thickness. The focal RPE bumpiness is shown below and nasally to the foveola (**C**). We report a 57-year-old woman who presented to an ophthalmologist in September 2022 with decreased visual acuity (VA) in her right eye. She complained of blurred vision and worsening of VA for the past 2 months. She has no history of ocular trauma or retinal or choroidal diseases. In 2014, she was diagnosed with breast cancer and underwent mastectomy with a one-month course of radiotherapy. In 2015, the patient started additional hormone therapy with tamoxifen 20 mg daily and has continued this therapy for 7 years without ophthalmological appointments during therapy. The patient underwent a complete ophthalmic examination, including slit lamp biomicroscopy with dilated fundus examination, best-corrected visual acuity (BCVA), refractive error measurement and intraocular pressure (IOP). BCVA was 0.5 in the RE and 0.8 in the left eye (LE), respectively (given in decimal acuity, measured with LogMAR charts). IOP was within normal limits. There were no abnormalities in the anterior segments of both eyes. Dilated fundus examination revealed uncharacteristic maculopathy in the RE and epiretinal membrane (ERM) in the LE. Multimodal imaging methods, including color fundus photography, fundus autofluorescence (FAF), optical coherence tomography (OCT) and OCT angiography (OCTA) were performed to confirm diagnosis of unilateral tamoxifen associated retinopathy (TAR) in the RE. The examination was performed using SS-OCT (DRI OCT Triton; Topcon, Tokyo, Japan).

**Figure 2 diagnostics-13-01250-f002:**
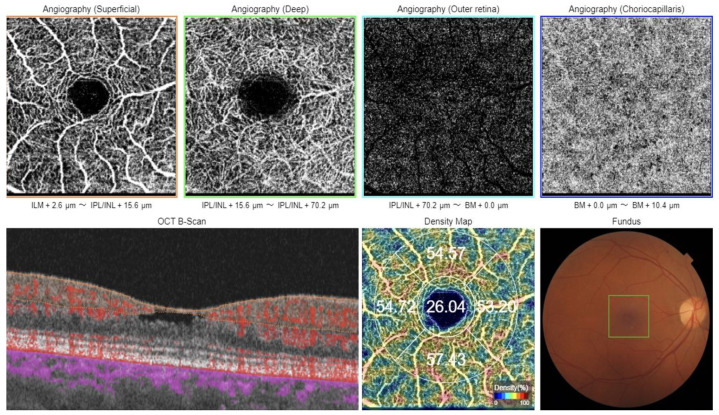
OCTA of the RE in patient with TAR reveals no abnormalities in the retinal circulation in both superficial and deep retinal plexuses and normal vessel density. Tamoxifen is the most common therapy for breast cancer in female [1,2,3,4,5,6,7]. It is an oral selective estrogen receptor modulator (SERM) that inhibits the division of neoplastic cells estrogen receptors (ER) positive forms of breast cancer [5]. Therefore, tamoxifen plays an important role in the therapy of more than two-thirds of postmenopausal breast cancer cases [1,2,3,4]. The standard dose of the drug is 20 mg once a day for 3 to 5 years. In more progressive cancer forms, the dose can be increased to 30–40 mg. Tamoxifen can also be used in malignant brain glioma treatment at high doses (200 mg/day). Use of the drug is associated with a low incidence of ophthalmological side-effects [7]. According to the available literature, less than 3% of patients were withdrawn from therapy due to its intolerance (mainly for gastrointestinal reasons and dizziness) [5]. Ophthalmic complications of tamoxifen therapy are relatively rare [7]. The range of ocular toxicity is between 0.9% and 12.0% and increases in higher drug doses [8]. Many studies revealed retinal pathologies, such as maculopathy and retinopathy, in cases of tamoxifen therapy. In the available literature, only some of the ocular complications were correlated with treatment duration. In the occurrence of ocular side-effects, such as glaucoma and macular degeneration, no correlation with the therapy time was reported [9,10]. In contrast to that, the incidence of cataracts due to 5-year tamoxifen therapy was significantly higher among long-term users (18.2% vs. 14.8%) [10]. The pathophysiology of retinal changes associated with tamoxifen is unclear. In 1978, Kaiser-Kupfer et al. indicated that crystalline retinal deposits formation in tamoxifen users can be related to axonal degeneration. The intracellular location of the retinal lesions in the nerve fiber and inner plexiform layers (IPL) were similar to nerve synapses seen in electron microscopy. The demonstration by histochemical methods of glycosaminoglycans in the deposits supported these changes [11]. Previously, it has been also postulated that tamoxifen binds with polar lipids and inhibits its normal catabolism. This results in the accumulation of drug-lipid complexes in lysosomes. It was shown that tamoxifen reduces the activities of lysosomal enzymes in retinal pigment epithelium (RPE) cells and indicated its toxicity to RPE in vitro [12]. In 2001, it was also noted that the phagocytosis of rod outer segments is inhibited by tamoxifen in RPE cell cultures [13]. Multiple case reports have shown that tamoxifen may cause crystalline deposits and pseudocystic foveal cavitations [11,14,15]. Crisostomo et al. described that patients who have been treated with this drug demonstrated reduced subfoveal choroidal thickness (CT) and total retinal thickness compared to the control group [2]. The authors suggested that there were structural changes in patients without symptoms, which could be an early sign of RPE and photoreceptor damage. To summarize, the macular changes after tamoxifen treatment range from severe crystalline retinopathy and cystoid macular edema (CME) with high doses of tamoxifen, to subtle macular crystal deposition and pseudocystic cavitary spaces visible only in OCT. In OCT, both TAR and type 2 macular telangiectasias (Mac Tel 2) can look similar. Visible changes in both of them include hyporeflective spaces in the fovea that first appear in the inner layers and may extend to the whole of the retina in the later stages of the disease.

**Figure 3 diagnostics-13-01250-f003:**
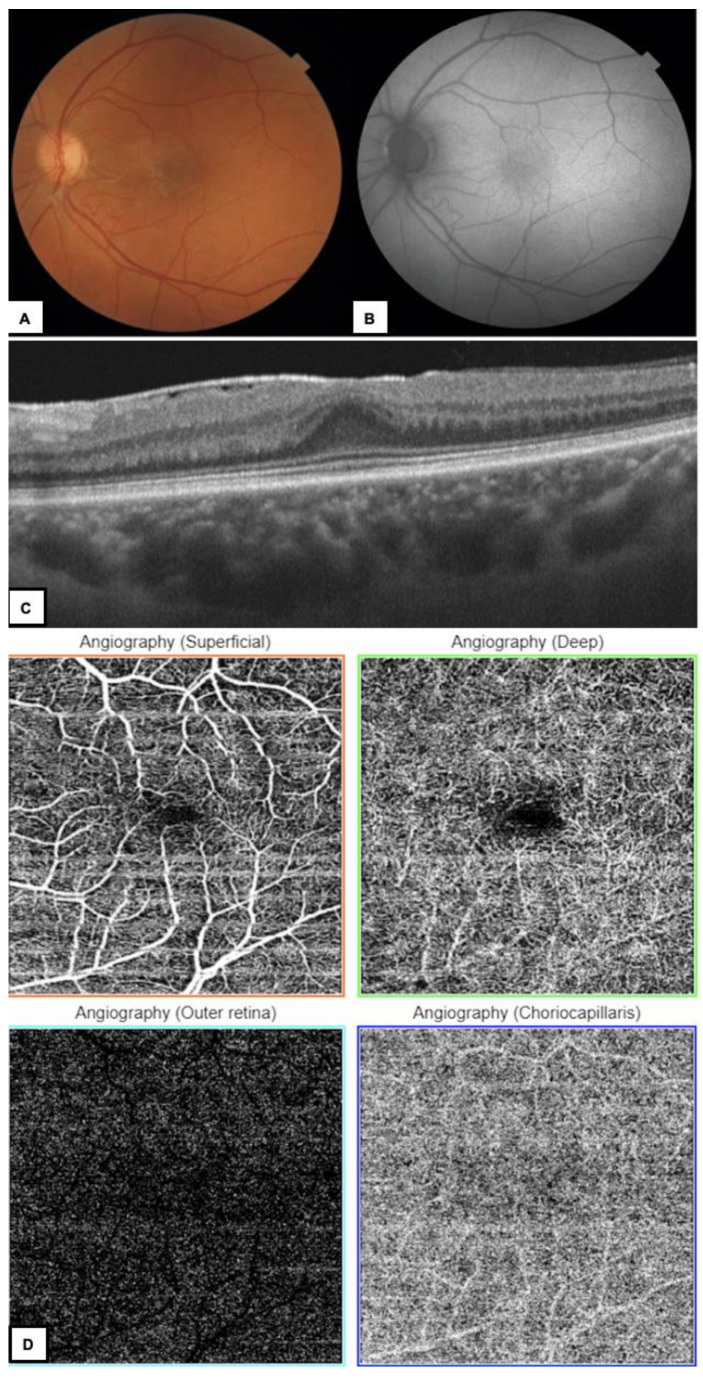
(**A**) Color fundus photography of the LE-wrinkling of the macular region due to ERM. (**B**) FAF–irregular autofluorescence in the macular area. (**C**) OCT B–scan shows ERM, shallow fovea with increased central retinal thickness. (**D**) OCTA–increased vascular tortuosity in the superficial retinal plexus and decreased foveal avascular zone area both in superficial and deep retinal plexuses due to ERM. OCTA was a conclusive imaging method providing the diagnosis of tamoxifen- induced maculopathy. The differential diagnosis of TAR is mainly Mac Tel 2 (Appendix A) [16,17,18]. The authors suggested a similar pathogenesis of TAR and Mac Tel 2 related to Muller cells dysfunction [19,20,21]. Mac Tel 2 can be seen in fundoscopy as a loss of foveal translucency, crystalline deposits and loss of macular pigment, but these abnormalities may not be clear enough to confirm the diagnosis. Available studies have proven the important role of OCTA in differentiating TAR from Mac Tel 2 [19,20,21]. The authors showed that in Mac Tel 2 OCTA reveals progressive capillary rarefaction, right angle venules, capillary ectasia in both superficial and deep perifoveal and parafoveal capillary networks [19,20,21]. In Todorich et al., study OCTA revealed normal superficial capillary plexus and flow voids in the parafoveal area and lower foveal vessel density in the deep plexus in a 53-year-old-woman with TAR [19]. Lee et al. found saccular capillary telangiectasias in the deep vascular plexus and in some patients with TAR and absence of these changes was interpreted as a mild form of retinopathy [20]. Hess et al., comparing patients with Mac Tel 2 and TAR, highlighted that in Mac Tel neurodegenerative and vascular alterations were equally present, while in TAR patients neurodegenerative changes predominated. In our case, OCTA showed no abnormalities in either superficial or deep capillary plexuses. Mac Tel 2 usually occurs bilaterally. Currently, there are no epidemiological data on how often TAR occurs uni- or bi-laterally. In the available literature, most cases occurred bilaterally [9,10,11,14,22]. The pseudocystic foveal cavitations noted in tamoxifen retinopathy can be differentiated from CME also by absence of leakage in fluorescein angiography (FA) and normal-to-reduced retinal thickness in OCT [16]. In our case, only non-invasive imaging methods were used. There were no clinical indications to perform additionally invasive FA. Due to the history of tamoxifen therapy, the unilateral appearance of the lesions, the characteristic OCT images and the lack of characteristic Mac Tel 2 changes in OCTA, the patient was diagnosed with tamoxifen-induced maculopathy. TAR should be considered in every visual disturbance among tamoxifen–users. Moreover, healthy women considering taking this drug to prevent breast cancer should be informed about its side-effects. Tamoxifen treatment can cause ocular complications during and after treatment. Recent studies have recommended a complete eye examination for baseline visual function parameters before initiating tamoxifen treatment [10]. Multimodal imaging, especially fundus photography and OCT helps in monitoring possible effects of its therapy. The correct diagnosis of TAR is very important in further therapeutic decisions in tamoxifen–users. Based on our diagnosis, the oncologist discontinued tamoxifen therapy. Despite the contradictory data available in the literature regarding the possibility of retinal lesions regression, the treatment method was switched in order to stop further progression of the disease [9,18]. The patient has been treated with letrozole 2.5 mg once a day since October 2022 and has regular ophthalmological examinations. Visual acuity, OCT and OCTA results remain stable.

## Data Availability

Data are available from the corresponding author upon reasonable request.

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
