# Peer review of "Unilateral Tamoxifen-Induced Retinopathy as a Consequence of Breast Cancer Treatment—Multimodal Imaging Value"

_diagnostics, 2023, doi:10.3390/diagnostics13071250_

Round 1

Reviewer 1 Report

Thank you for a nice paper summarising the findings that can be found in tamoxifen induced retinopathy. These findings are not a new discovery and it would be helpful to know what is the current outcome of your patient since the tamoxifen has stopped and whether the changes on the OCT or OCTA has changed or the vision and symptoms of the patient has improved. It will be helpful to know if the literature have any information on progression or regression of the condition with or without stopping the tamoxifen.

Also it will beneficial in the paper to highlight where the figures are related to in the main text manuscript.

Maybe an image of OCT or OCTA with someone with Mac Tel can help the reader see the difference rather than just a description on the manuscript.

Overall, the paper does not add anything extra to the literature but it will be useful information for a non ophthalmologist to be aware of these changes that can happen in patient using tamoxifen.

Besides that, please note of the grammatical errors that need to be corrected and clarity of the sentences to make the paper flow a bit better:

Line 13: change increasing to higher

Line 13: Rephrase to “A rare known complication of tamoxifen use is the development of retinopathy”

Line 16: Change “what allowed to” à “helps confirm the diagnosis of”

Line 17: Remove “Among the multimodal imaging methods”

Line 19: Change “mainly in the differentiation” à especially in differentiating it

Line 20: Change “further therapeutic decisions” à deciding the treatment option of

Line 21: Change “another treatment” à another course of treatment before stopping the tamoxifen therapy completely.

Line 22: Change this sentence to describe what is happening to the patient now.

Line 33: RE. Please describe it as right eye (RE).

Line 34: Change “couple” à the past 2 months

Line 34: Change “had” à has

Line 34: Remove “and had nerve been treated for ophthalmological reasons”

Line 38: Rephrase to mean that patient not had ophthalmology review until reduction of vision.

Line 42: What type of measurement used for the 5/10 in RE and 5/6 in LE. Is this Snellen? LogMar? etc.

Line 43: LE. Please describe it as left eye (LE)

Line 46: Please describe FAF as fundus autofluorescence (FAF) and OCT as optical coherence tomography (OCT).

Line 49: Change “is recommended … breast cancer” à is the most common therapy for breast cancer in females.

Line 51: Remove the words “with many” and “in ER”.

Line 56: Change to 200mg/day (not 200mg/d)

Line 56-58: Justify the low incidence of side effects with a reference and what intolerance is found?

Line 62: Not sure what it means in this sentence regarding glaucoma and macular degeneration did not depend on the treatment duration. Need to rephrase this.

Line 64-67: Need to combined the sentence and rephrase as it is quite repetitive.

Line 69: Change “drug users” à tamoxifen users

Line 70-73: Again very repetitive sentence and need to rephrase.

Line 74: Change “inhibiting” à inhibits

Line 74: Change “that mechanism” à this

Line 75: RPE to describe as retinal pigment epithelium (RPE).

Line 76: Change “and indicated … in vitro” à and toxicity to the RPE cells in vitro.

Line 87: Remove “like in our patient”

Line 89: Change “entire retina in later stages..” à whole of the retina in the later

Line 96: Change “concerns” à is

Line 98: Remove “IN the fundus examination”

Line 108-109: Replace whole sentence as very repetitive and confusing as it says may occur in one or both eyes repetitively.

Line 126: Change “complete” à “a complete”

Line 127: Change “to determine visual function” à “for baseline visual function parameters”

Line 127: Change “multimodal imaging” to describe the type of multimodal imaging that you will recommend to use (eg. OCT, OCTA, FA etc)

Line 133: Rephrase last sentence as it would be good to know what is happening to the patient’s current vision and changes on imaging.

Author Response

Dear Reviewer, 

Thank you for your constructive and informative review of our paper. We have answered each of your points below.

Thank you for a nice paper summarising the findings that can be found in tamoxifen induced retinopathy. These findings are not a new discovery and it would be helpful to know what is the current outcome of your patient since the tamoxifen has stopped and whether the changes on the OCT or OCTA has changed or the vision and symptoms of the patient has improved. It will be helpful to know if the literature have any information on progression or regression of the condition with or without stopping the tamoxifen.

We have completed the text about information on current outcome of our patient. In the available literature, we found one report with information that ocular complications of tamoxifen seldom cause significant visual impairment and, except for crystalline retinopathy, are reversible upon discontinuation of tamoxifen.

Also it will beneficial in the paper to highlight where the figures are related to in the main text manuscript.

The figures are related to the text below, we have constructed our case as an “Interesting Images” article following specific instructions for Authors.

Maybe an image of OCT or OCTA with someone with Mac Tel can help the reader see the difference rather than just a description on the manuscript.

We have added an extra Figure with Mac Tel 2 to the supplementary materials.

Overall, the paper does not add anything extra to the literature but it will be useful information for a non ophthalmologist to be aware of these changes that can happen in patient using tamoxifen.

Thank you for the comment. Describing our case, we would like to emphasize that the correct diagnosis of TAR is very important in further therapeutic decisions in patients treated with tamoxifen.

We have corrected the errors according to your suggestions:

Besides that, please note of the grammatical errors that need to be corrected and clarity of the sentences to make the paper flow a bit better:

Line 13: change increasing to higher

Line 13: Rephrase to “A rare known complication of tamoxifen use is the development of retinopathy”

Line 16: Change “what allowed to” à “helps confirm the diagnosis of”

Line 17: Remove “Among the multimodal imaging methods”

Line 19: Change “mainly in the differentiation” à especially in differentiating it +

Line 20: Change “further therapeutic decisions” à deciding the treatment option of

Line 21: Change “another treatment” à another course of treatment before stopping the tamoxifen therapy completely.

Line 22: Change this sentence to describe what is happening to the patient now.

Line 33: RE. Please describe it as right eye (RE).

Line 34: Change “couple” à the past 2 months

Line 34: Change “had” à has

Line 34: Remove “and had nerve been treated for ophthalmological reasons”

Line 38: Rephrase to mean that patient not had ophthalmology review until reduction of vision.

Line 42: What type of measurement used for the 5/10 in RE and 5/6 in LE. Is this Snellen? LogMar? etc.

We completed the text with information about the  type of VA measurement.

Line 43: LE. Please describe it as left eye (LE)

Line 46: Please describe FAF as fundus autofluorescence (FAF) and OCT as optical coherence tomography (OCT).

Line 49: Change “is recommended … breast cancer” à is the most common therapy for breast cancer in females.

Line 51: Remove the words “with many” and “in ER”.

Line 56: Change to 200mg/day (not 200mg/d)

Line 56-58: Justify the low incidence of side effects with a reference and what intolerance is found?

Low incidence refers to ophthalmological side-effects (reference number 7), we have corrected this.

We have quoted a source citing a paper entitled “Review of the clinical pharmacology and international experience with tamoxifen in advanced breast cancer” by Patterson JS, Battersby LA, Edwards DG. It is difficult to determine which population the study was conducted on, as Patterson et al. summarized 12 major trials on tamoxifen. They found 27 withdrawals from therapy in 988 patients (3%)– mainly for gastrointestinal reasons and dizziness (not for ocular toxicity)– we have supplemented the text with this information. We have also corrected error in this citation in the references.

Line 62: Not sure what it means in this sentence regarding glaucoma and macular degeneration did not depend on the treatment duration. Need to rephrase this.

We have described it in the manuscript.

Line 64-67: Need to combined the sentence and rephrase as it is quite repetitive.

We have combined the sentence.

Line 69: Change “drug users” à tamoxifen users

Line 70-73: Again very repetitive sentence and need to rephrase.

We have rephrased it.

Line 74: Change “inhibiting” à inhibits

Line 74: Change “that mechanism” à this

Line 75: RPE to describe as retinal pigment epithelium (RPE).

Line 76: Change “and indicated … in vitro” à and toxicity to the RPE cells in vitro.

Line 87: Remove “like in our patient”

Line 89: Change “entire retina in later stages..” à whole of the retina in the later

Line 96: Change “concerns” à is

Line 98: Remove “IN the fundus examination”

Line 108-109: Replace whole sentence as very repetitive and confusing as it says may occur in one or both eyes repetitively.

We have rephrased it.

Line 126: Change “complete” à “a complete”

Line 127: Change “to determine visual function” à “for baseline visual function parameters”

Line 127: Change “multimodal imaging” to describe the type of multimodal imaging that you will recommend to use (eg. OCT, OCTA, FA etc)

Line 133: Rephrase last sentence as it would be good to know what is happening to the patient’s current vision and changes on imaging.

We have completed the text with information about current vision acuity and OCT and OCTA results.

Reviewer 2 Report

1.      Abstract:

a.      Line 10: instead of ‘with estrogen receptors’, suggest using perhaps ‘targeting estrogen receptors’ or something like that.

b.      Similarly, line 16, ‘for’ instead of ‘due to’

2.      Figures:

a.      Figure 1: It will be helpful to show the orientation/ location of the OCT scan on the image.

b.      Line 31: there is also a small bumpy RPE in figure 1D.

c.      Figure 2: The angiography images, especially the deep capillary plexus, don’t look very clear at the foveal region, so it is hard to tell whether there was any abnormality or not.

d.      It will be informative to include what devices were used for OCT, OCTA, etc.

3.      Line 42: it will be helpful to mention what visual acuity notation this is. If it is decimal acuity, the authors can mention that.

4.      Line 57: I would ask for a little more detail about these incidence data. 3% of patients worldwide, or in Poland, or wherever the reference 5 study was done.

5.      Line 60: Again, I would be careful about interpreting and describing these incidence data. In the abstract, it was mentioned as incidence of ocular toxicity. However, here, it is described as the range of ocular toxicity. Kindly correct the description.

6.      Line 64: which authors described about the frequency of cataracts. Please include reference.

7.      Line 68 to 78: it will be easy to read if the authors focused on describing the pathophysiology as a summary and giving the reference rather than focusing on the authors who said it

8.      Line 89 – is this description related to Mac Tel 2 or TAR. I think it should be clarified.

9.      Line 100 - ?available studies. Please add references.

10.   I just looked up the literature and found some studies (Hess, K, 2023, Ophthalmology retina; Kovach, 2020, Ann Eye Sci) that indicate OCTA changes in TAR similar to that of MacTel 2. The authors should refer to them and look closely at the OCTA images at a better resolution to make sure that there were no abnormalities. Or give a possible explanation on why they think there were no abnormalities when other studies have shown them. This could be a unique case report for the reason that there are no OCTA abnormalities.

Author Response

Dear Reviewer, 

Thank you for your constructive and informative review of our paper. We have answered each of the points below.

  1. Abstract:
  2. Line 10: instead of ‘with estrogen receptors’, suggest using perhaps ‘targeting estrogen receptors’ or something like that.

We have changed this phrase.

  1. Similarly, line 16, ‘for’ instead of ‘due to’

We have corrected this error.

  1. Figures:
  2. Figure 1: It will be helpful to show the orientation/ location of the OCT scan on the image.

According to your suggestion, we have supplemented Figure 1 with location of the OCT scan.

  1. Line 31: there is also a small bumpy RPE in figure 1D.

We have completed the description of Figure 1d.

  1. Figure 2: The angiography images, especially the deep capillary plexus, don’t look very clear at the foveal region, so it is hard to tell whether there was any abnormality or not.

We have completed the description of the OCTA images in Figure 2.

  1. It will be informative to include what devices were used for OCT, OCTA, etc.

We have included this information.

  1. Line 42: it will be helpful to mention what visual acuity notation this is. If it is decimal acuity, the authors can mention that.

We have completed the text with this  information.

  1. Line 57: I would ask for a little more detail about these incidence data. 3% of patients worldwide, or in Poland, or wherever the reference 5 study was done.

We have quoted a source citing a paper entitled “Review of the clinical pharmacology and international experience with tamoxifen in advanced breast cancer” by Patterson JS, Battersby LA, Edwards DG.”. It is difficult to determine which population the study was conducted on, as Patterson et al. summarized 12 major trials on tamoxifen. They found 27 withdrawals from therapy in 988 patients (3%)– mainly for gastrointestinal reasons and dizziness (not for ocular toxicity) – we have supplemented the text with this information. We have also corrected error in this citation in the references.

  1. Line 60: Again, I would be careful about interpreting and describing these incidence data. In the abstract, it was mentioned as incidence of ocular toxicity. However, here, it is described as the range of ocular toxicity. Kindly correct the description.

We have corrected this in the manuscript main body - the range of ocular toxicity is between 0.9% and 12.0% and increases in higher drug doses.

  1. Line 64: which authors described about the frequency of cataracts. Please include reference.

It was the same source as in the next sentence - reference number 10. We have already added it to the both sentences.

  1. Line 68 to 78: it will be easy to read if the authors focused on describing the pathophysiology as a summary and giving the reference rather than focusing on the authors who said it

We have already corrected this.

  1. Line 89 – is this description related to Mac Tel 2 or TAR. I think it should be clarified.

It was related to both, MacTel2 and TAR. We have already clarified it in the text.

  1. Line 100 - ?available studies. Please add references.

It was the same source as in the next sentence - reference number 19. We have already added it to both sentences.

  1. I just looked up the literature and found some studies (Hess, K, 2023, Ophthalmology retina; Kovach, 2020, Ann Eye Sci) that indicate OCTA changes in TAR similar to that of MacTel 2. The authors should refer to them and look closely at the OCTA images at a better resolution to make sure that there were no abnormalities. Or give a possible explanation on why they think there were no abnormalities when other studies have shown them. This could be a unique case report for the reason that there are no OCTA abnormalities.

Mild alterations in OCTA were reported by Todorich et al. in a 53-year-old woman with a history of breast cancer, placed on tamoxifen 20 mg daily. The OCTA illustrated only flow voids in the temporal parafoveal deep capillary plexus, the rest of the results of OCTA were normal. In our case there were no abnormalities. We also completed the references with suggested articles.

Round 2

Reviewer 1 Report

Better read now after changes.

Please re-edit the sentence below in line 81 to the following:

The intracellular location of the retinal lesions in the nerve fiber and inner plexiform layers (IPL) were similar to nerve synapses seen in electron microscopy. The demonstration by histochemical methods of glycosaminoglycans in the deposits supported these changes [11].

Author Response

Dear Reviewer, 

Thank you for review of our paper. We have answered each of your points below.

The intracellular location of the retinal lesions in the nerve fiber and inner plexiform layers (IPL) were similar to nerve synapses seen in electron microscopy. The demonstration by histochemical methods of glycosaminoglycans in the deposits supported these changes [11].

We have re-edited this sentence. 

Best regards,

Paulina Szabelska